# Cytogenetic, Serum Liver Enzymes and Liver Cell Pathology of the Hampala Barb Fish *(Hampala macrolepidota*) Affected by Toxic Elements in the Contaminated Nam Kok River near the Sepon Gold-Copper Mine, Lao PDR

**DOI:** 10.3390/ijerph18115854

**Published:** 2021-05-29

**Authors:** Latsamy Soulivongsa, Bundit Tengjaroenkul, Isara Patawang, Lamyai Neeratanaphan

**Affiliations:** 1Toxic Substances, Microorganisms and Feed Additives in Livestock and Aquatic Animals for Food Safety Research Program, Khon Kaen University, Khon Kaen 40002, Thailand; soulivongmee@gmail.com (L.S.); btengjar@kku.ac.th (B.T.); 2Division of Livestock Medicine, Faculty of Veterinary Medicine, Khon Kaen University, Khon Kaen 40002, Thailand; 3Research Center in Bioresources for Agriculture, Industry and Medicine, Chiang Mai University, Chiang Mai 50200, Thailand; isara.p@cmu.ac.th; 4Division of Environmental Science, Faculty of Science, Khon Kaen University, Khon Kaen 40002, Thailand

**Keywords:** chromosome, environment, fish, lesion, metal, toxicity

## Abstract

This study aimed to determine toxic element concentrations in aquatic environments, including water and sediment, and in the *Hampala macrolepidota* fish, and to evaluate chromosome abnormalities, serum liver enzyme changes and liver histopathological alterations in *H. macrolepidota* from the Nam Kok River near the Sepon gold-copper mine, Lao People’s Democratic Republic, as compared with a control area without mining activity. The results revealed significant differences (*p* < 0.05) in As, Ba, Cu, Fe, Mn, Ni, Se and Zn in water, in all of the studied potentially toxic elements in sediment, and in As, Ba, Cd, Cr, Cu, Mn, Ni, Se, and Zn in the fish between the study and control areas. A chromosome assessment demonstrated 6 types of chromosome abnormalities, among which centric gap had the highest total number of chromosome abnormalities. Percentage of chromosome abnormalities, percentage of cells with chromosome abnormalities and serum liver enzymes in *H. macrolepidota* were significantly different (*p* < 0.05) between the two studied areas and were higher in the contaminated fish than in the control fish. The observation of liver histopathological changes revealed cellular degeneration, such as nuclear damage, abnormal cytoplasmic mitochondria and the disintegration of rough endoplasmic reticulum. The results indicate that the contamination of potentially toxic elements in the Nam Kok River near the Sepon gold-copper mine area negatively affected chromosomes, serum liver enzymes and liver cell structures in *H. macrolepidota*.

## 1. Introduction

The Lao People’s Democratic Republic includes mountainous regions with natural scenery as well as various valuable mineral resources in several regions of the country. To date, the Lao government has promoted ore resources as a mechanism for increasing the gross domestic product of the country [1,2]. In 2018, the major valuable ore commodities from the Lao People’s Democratic Republic were gold and copper, with estimated export values of 200 and 502 million USD, respectively [3,4]. By 2020, the value of the total metal exports is predicted to account for 10% of Laos’s gross domestic product. The Sepon gold-copper mine in the Savannakhet Province is the second largest gold-copper mine, which contributes to 90% of the country’s total economic value from the mining sector [1,3]. In addition to the mining industry, aquaculture and fishery are major essential economic sectors in the Lao People’s Democratic Republic, accounting for more than half of animal protein sources [5]. However, at present the volume of fish captured from streams, rivers and reservoirs in Lao PDR is decreasing due to the low surface water qualities, which is similar to other polluted areas [6,7]. Previous reports have mentioned that potentially toxic elements can negatively affect the structure and metabolism of animal and human cells, tissues, and organs, can cause chromosome and DNA damage, tumors, and can drive evolution [8,9,10].

*Hampala**macrolepidota* is a carnivorous fish species with a high demand for human consumption in Laos and other countries in Southeast Asia [11]. Fish are generally accepted as the main depots of potentially toxic elements in ecosystems, and they are selected as appropriate biomonitors for the assessment of potentially toxic element contamination in aqueous environments [12,13,14]. As a result, the rapid growth of gold-copper mining in the Lao People’s Democratic Republic has raised intense concerns involving potentially toxic element emissions into aquatic ecosystems; however, data involving types and concentrations of potentially toxic elements in the environment and the health effects of these elements on *H**. macrolepidota* are limited. Only Soulivongsa et al. [15] has reported on five toxic elements in the Nam Kok River and aquatic organisms. Fe, Mn and Ni in the river water as well as As and Cd in the sediment and As and Cd in *Osterochilus vittatus* fish samples in the river exceeded standard values. Therefore, studies on the concentrations of potentially toxic elements, including As, Cd, Pb, Cr, Ba, Cu, Fe, Mn, Ni, Se, and Zn in water, sediment and *H**. macrolepidota*, as well as their potential toxic effects on chromosomes, serum liver enzyme indices and the liver cell pathology of *H**. macrolepidota* in the Nam Kok River were conducted. Information on the potentially toxic elements impacting several aspects of this carnivorous fish species can be applied for environmental and human health risk management as well as for setting regulations, laws and policies for the mining industry in the Lao People’s Democratic Republic.

## 2. Materials and Methods

### 2.1. Study Area

The Nam Kok River, located near a Sepon gold-copper mine in the Viraboury District, Savannakhet Province, Lao People’s Democratic Republic, at GPS location 16°52′06.32′′ N 106°02′22.88′′ E, was set as the study area, and the Nam Souang River in the Naxaythong District, Vientiane, at GPS location 18°14′52.88′′ N 102°27′54.78′′ E, near agricultural and domestic areas without mining and electronic activities, i.e., potential load for naturally leached elements, was set as a control area (Figure 1).

### 2.2. Water Quality Parameters

The water quality parameters, including dissolved oxygen (DO), hydrogen potential (pH), temperature, total hardness (TH), carbonate hardness (CH) and electrical conductivity (EC), were determined at the experimental areas using electrical detectors at 8:00 a.m., whereas nitrite, nitrate and ammonia were determined using standard titration methods (Table 1).

### 2.3. Sample Collection

Water, sediment, and *H**. macrolepidota* fish samples were randomly collected in three replicates (n = 9) from the Nam Kok River near the Sepon gold-copper mine (as the study area) and the Nam Souang River (as the control area) during October–December 2019 (Figure 1). In order to avoid resuspension concern, the water samples were collected at a depth of 30 cm from the surface before sediment collection. Nitric acid was then added to the water samples, and the sediment samples were air dried before quantification of potentially toxic element concentrations. Ten *H**. macrolepidota* samples were measured for potentially toxic element contents in the fish muscles as a result of biomagnification in the food chain due to the risks to human health, including chromosome abnormalities from kidney cells, serum liver enzymes and liver histopathological changes. The average body weights of *H**. macrolepidota* taken from the Nam Kok and Nam Souang Rivers were not significantly different (*p* > 0.05), at approximately 103.00 ± 13.82 g and 97.20 ± 9.08 g, respectively.

### 2.4. Measurements of the Potentially Toxic Element Concentrations

#### 2.4.1. Water Samples

A total of 25 mL of each water sample was mixed with 1.25 mL of 30% HNO_3_ (ACS grade), then the sample was digested on a hot plate at 95 ± 5 °C for 60 min. After cooling, the digested sample was adjusted to 25 mL with deionized water and filtered through cellulose paper No. 1. The final sample was analyzed via inductively coupled plasma optical emission spectrometry (ICP-OES) [16].

#### 2.4.2. Sediment Samples

Each sediment sample with an exact weight of 1.0 g (Mettler Toledo, Model MS 105, Greifensee, Switzerland) was sequentially digested with 5 mL of nitric acid (ACS grade), 15 mL of hydrochloric acid (ACS grade) and 10 mL of hydrogen peroxide (ACS grade) in a heating mantle at 95 ± 5 °C for 2 h. After cooling, the sample was adjusted to a volume of 50 mL with deionized water and filtered through cellulose filter paper No. 42. The final samples were then analyzed via ICP-OES (PerkinElmer, Inc., Model optima 8300, Waltham, MA, USA) [17].

#### 2.4.3. Fish Samples

Each fish muscle sample with an exact weight of 1.0 g was mixed with sulfuric and nitric acid (ACS grade) and digested on a hot plate at 60 °C for 30 min. Then, 10 mL of hydrogen peroxide (ACS grade) was mixed and digested on a hot plate for 1 h. After cooling, the digested sample was adjusted to a volume of 25 mL with deionized water. Then the mixture was passed through filtered paper No. 1. Finally, the sample was analyzed via ICP-OES (PerkinElmer, Inc., Model optima 8300, Waltham, MA, USA) [16].

#### 2.4.4. Quality Control and Quality Assurance

The ICP-OES wavelengths for the analysis of As, Ba, Cd, Cr, Cu, Fe, Mn, Ni, Pb, Se and Zn were set as 188.979, 233.527, 226.502, 267.716, 324.752, 259.939, 259.327, 231.604, 220.353, 196.022, and 213.857 nm respectively, and the detection limits of the analyzed elements were As:0.006, Ba:0.002, Cd:0.001, Cr:0.001, Cu:0.002, Fe:0.002, Mn:0.002, Ni:0.001, Pb:0.005, Se:0.023, and Zn:0.001 mg/L. Analyses of blanks and standards were conducted for every 10th sample. The concentrations of the potentially toxic elements in the procedural blanks were significantly less than 5% of the mean analyzed concentrations for all potentially toxic elements. Replications of the analyses were conducted to guarantee the precision and accuracy of all measurements. The results were not found to deviate by more than 2% from the certified levels. The potentially toxic element recovery values were then calculated through acceptance in the range of 85–115%. The values considered accurate were 90–100% [18,19].

### 2.5. Chromosome Preparation and Assessment

*H**. macrolepidota* were injected with colchicine at 0.05% *v*/*w* and left for 1 h. Then, the kidney was cut into small pieces, mixed with 8 mL of 0.075 M KCl (ACS grade) for 25 min and centrifuged at 1500 rpm for 10 min. The kidney cells were then fixed in a cool fixative (3 methanol:1 glacial acetic acid). The fixative was gradually added up to 8 mL before centrifugation at 1500 rpm for 10 min. The fixation process was repeated as the supernatant was cleared. The sediment was mixed with 1 mL of fixative, and the cell suspension was dropped on a glass slide. The air-dried slide was stained with a 20% Giemsa’s solution for 30 min [20,21]. A total of 500 metaphase chromosomes were photographed, counted and recorded under a light microscope at a magnification of 1000×. The total number of chromosome abnormalities, total number of cells with chromosome abnormalities and the percentage of chromosome abnormalities were calculated and analyzed statistically.

### 2.6. Serum Liver Enzymes

Clotted blood samples from the *H**. macrolepidota* caudal vessel were centrifuged at 2000 rpm for 10 min, and the fish serum was stored at −20 °C before liver enzyme determination [22]. Serum enzymes aspartate aminotransferase (AST) and alanine aminotransferase (ALT) were measured using an automated analyzer (ROCHE/Hitachi Cobas C501, ROCHE/Hitachi, Roche Diagnostics K.K., Tokyo, Japan).

### 2.7. Liver Histopathology Study

*H**. macrolepidota* livers were fixed in 1% osmium tetroxide (ACS grade) for 1 day, dehydrated in a series of ethyl alcohols (ACS grade), transferred to propylene oxide (ACS grade), and embedded in capsules with an Eppon polymer. Sections cut ultrathin sections using an ultramicrotome were stained with uranyl acetate and lead citrate (ACS grade) and observed under a transmission electron microscope (JEOL 100, JEOL Ltd., Tokyo, Japan) [23].

### 2.8. Statistical Analysis

The potentially toxic element concentrations in contaminated water, sediment, and fish muscles as well as liver enzymes were tested with independent t-tests comparing the differences between two unrelated areas. Water quality parameters, the total number of chromosome abnormalities, the cell numbers with chromosome abnormalities and the percentage of chromosome abnormalities in each fish near the Sepon gold-copper mine and the control areas were compared and analyzed statistically as independent observations using a Mann–Whitney U-test comparing between two independent areas without normal distribution, with SPSS program version 24, at a 95% confidence level. Histological changes in fish liver cells were descriptively reported.

## 3. Results

### 3.1. Parameters of Water Quality

The values of the water quality parameters of water samples from the Nam Kok River near the gold-copper mine (as the study area) and the Nam Souang River (as the control area) are shown in Table 2.

### 3.2. Potentially Toxic Element Concentrations in Water Samples

The average potentially toxic element concentrations in the water samples as mg/L are shown in Table 3. There were significant differences (*p* < 0.05) in the concentrations of all metals between the study and control areas. The average potentially toxic element concentrations of Cd, Cr, Fe, Mn and Pb in the water samples in the study and control areas were 0.002 ± 0.001 and 0.001 ± 0.001 mg/L; 0.017 ± 0.007 and 0.018 ± 0.015 mg/L; 0.638 ± 0.126 and 0.910 ± 0.265 mg/L; 0.109 ± 0.039 and 0.112 ± 0.016 mg/L; and 0.038 ± 0.030 and 0.023 ± 0.019 mg/L, respectively. The results suggested that there were no significant differences (*p* > 0.05) in element concentrations other than for the As, Ba, Cu, Ni, Se and Zn in the water samples in both areas. The Cu and Se concentrations did not have reported standard values (Table 3).

### 3.3. Potentially Toxic Element Concentrations in Sediment Samples

The average potentially toxic element concentrations in sediment samples are shown as mg/kg in Table 4. The statistical results suggested that there were significant differences in all the studied potentially toxic element concentrations in sediment samples between the study area and the control area (*p* < 0.05). The concentrations of Ba, Fe and Se did not have reported standard values, and the concentration of Se in sediment in the Nam Kok area was not detected.

### 3.4. Potentially Toxic Element Concentrations in H. macrolepidota Samples

The average concentrations of potentially toxic elements in fish muscles as mg/kg from the Nam Kok River near the gold-copper mine (as the study area) and the Nam Souang River (as the control area) are shown in Table 5. Potentially toxic element concentrations accumulated in the fish muscles from both studied areas were statistically significant (*p* < 0.05), whereas the concentrations of Fe in fish muscles from both areas, 44.74 ± 19.21 and 39.33 ± 8.84 mg/kg, respectively, were not significantly different (*p* > 0.05). The concentrations of Ba, Fe, Ni, Se and Zn did not have reported standard values, and the concentration of Pb in the Nam Kok area was not detected (Table 5).

### 3.5. Chromosome Sssessment in H. macrolepidota

The diploid chromosome number (2n) of *H**. macrolepidota* from the study and the control areas was 2n = 50. The normal karyotype of *H**. macrolepidota* from both areas consisted of 3 pairs of metacentric, 9 pairs of submetacentric, 6 pairs of acrocentric, and 7 pairs of telocentric regions (Figure 2). Figure 3 shows the different types of chromosome abnormalities in the metaphase spread cells. The results revealed that the 6 types of chromosome abnormalities in *H**. macrolepidota* from the gold mining area were centric gaps, fragmentations, single chromatid gaps, deletions, single chromatid breaks and centric fragmentations, with a total number of chromosomal abnormalities being 127, 3, 3, 22, 1, and 15, respectively. The total number of chromosome abnormalities, cell numbers with chromosome abnormalities and the percentage of chromosome abnormalities of *H**. macrolepidota* in the study and the control areas were 171, 73 and 29.20% and 39, 27 and 10.80%, respectively. The statistical results indicated that the total number of chromosome abnormalities, the total cell numbers with chromosome abnormalities and the percentages of chromosome abnormalities from both areas were significantly different (*p* < 0.05) (Table 6).

### 3.6. Serum Liver Enzymes

The liver enzymes (AST and ALT) of the fish from the study area near the gold-copper mine were higher and significantly different from those of the fish in the control area (*p* < 0.05) (Table 7).

### 3.7. Liver Histopathology

The liver cells of the control *H. macrolepidota* fish showed normal structures, such as (1) oval nuclei with condensed nucleoli, (2) cell membranes that were smooth without broken boundaries, (3) rough endoplasmic reticulum arranged in parallel and (4) normal-sized mitochondria (Figure 4a). In contrast, abnormal liver cells of the fish exposed to toxic elements showed (1) irregularly-shaped nuclei with scattered and lighter-colored nucleoli, (2) nuclear membrane breakage, (3) rough endoplasmic reticulum disintegration and (4) swollen and lighter-colored mitochondria (Figure 4b).

## 4. Discussion

### 4.1. Water Quality Parameters

Most of the measurements of water quality parameters were within the standards for non-contaminated surface water [38], especially dissolved oxygen (DO), temperature, the potential of hydrogen (pH) and electrical conductivity (EC), and indicated that the water quality profile of the control area could be appropriate for fish to live, as recorded by Boyd [39]; Pruszynski [40]; Tram et al. [41]; Akinwole et al. [42]; Culioli et al. [43]; and Nguyen [44].

### 4.2. Potentially Toxic Element Concentrations in Water, Sediment and H. macrolepidota

The concentrations of As, Ba, Cu, Ni, Se and Zn in the water samples from the Nam Kok River near the gold-copper mine area and the control area were significantly different (*p* < 0.05), whereas the concentrations of Cd, Cr, Fe, Mn and Pb in the water from both of the studied areas were not significantly different (*p* > 0.05). However, the concentrations of Fe, Mn, Ni and Zn in the water from the Nam Kok River exceeded the water standards limit according to the FAO [26], FAO/WHO [27], US-EPA [28] and UNEPGEMS [29] notifications, as shown in Table 3. Furthermore, the Nam Souang River, as the control area, contains potentially toxic elements from agriculture and domestic activities, but does not exceed the water standards limit, except for Fe, Mn and Zn. This could imply that the studied river was naturally enriched with some elements in the collected water.

These results are in accordance with other studies. Intamat et al. [45] and Khamlerd et al. [46] reported that the concentrations of As, Cr, Fe, Mn, and Zn in water around the gold mining area of Loei Province and the Bueng Jode reservoir near an industrial area in Thailand were higher than the standard values. Additionally, the concentrations of As, Cd and Mn from other gold mines, lakes, reservoirs and domestic wastewater canals in Thailand were greater than the standard values and non-contaminated areas [16,46,47,48,49].

Most of the potentially toxic element concentrations in the sediment from both studied areas suggested that they were significantly different (*p* < 0.05). The average concentrations of As, Cd, Cu and Zn in the sediment samples of the Nam Kok River were greater than the standard values according to the standards of TPCD [31], TPCD [32] and the EU [33], while the concentrations of Cr, Ni, Mn and Pb in the sediment were lower than the standard values. When compared, the concentrations of potentially toxic elements As and Cd in the sediment of the Nam Kok River near the gold-copper-mine area were higher than those of other research studies

Intamat et al. [45] and Tengjaroenkul et al. [17] determined that As and Cd were the two most prevalent elements in the sediment around the gold mine area of Loei Province, Thailand, and that the concentrations of As, Cd, Cu, Cr, Pb, Ni, Fe, Zn and Mn reported in the Nam Kok River in this study were lower than those of other studies [45,46,50].

The average potentially toxic element concentrations in the muscle of *H**. macrolepidota* are shown in Table 5. In addition to Pb, ten potentially toxic element concentrations in the fish muscle samples from the study and the control areas were statistically significantly different (*p* < 0.05). The concentrations of Cr, Cd and Mn in *H**. macrolepidota* muscles in the study area exceeded the standard values for freshwater fish [35,36]. As a carnivorous fish, *H**. macrolepidota* can absorb potentially toxic elements into its body after feeding on contaminated elements via the digestive tract, which are directly distributed from toxic element-contaminated aquatic ecosystems into *H**. macrolepidota* gills and skin. From previous studies on the Nam Kok River, potentially toxic element accumulations were found to differ among fish species. Soulivongsa et al. [15] reported that bony lip barb fish (*Osteochilus vittatus*) contained As, Cr, Fe, Mn and Ni at greater concentrations than hampala barb fish in this study, with values of 3.48 > 0.30, 4.72 > 3.58, 56.10 > 44.74, 14.76 > 4.19 and 1.76 > 1.70 mg/kg, respectively. These differences suggested that bony lip barb fish, an herbivorous fish, may consume an abundance of contaminated aquatic plants in the river in greater amounts. Therefore, higher accumulated concentrations were observed at a greater level. Furthermore, Jiang et al. [49] found that potentially toxic elements in the muscles of *Cyprinus*
*carpio* and *C**. auratus* from Lake Caizi in Southeast China had higher concentrations than the levels near the gold-copper mine area in this study. Keshavarzi et al. [51] performed health risk assessments in three fish species (*Anodontostoma chacunda, Johnius belangerii* and *Cynogloddurs arel*) in the Persian Gulf in Iran and found that the concentrations of As and Cd were higher than those of the metals in the *H**. macrolepidota* muscle samples in this study. Furthermore, Neeratanaphan et al. [52] determined that the concentration of As in the *Monopterus albus* muscle and the concentrations of Cd, Cr, Cu, Ni and Zn in *Oreochromis niloticus* muscle [50] were greater than the metal levels in this study, and Keshavarzi et al. [51] detected As and Cd in *Anodontostoma chacunda*. Khamlerd et al. [46] reported the concentrations of Cr, Cd, As, and Ni in *Chana stariata* at Bueng Jode reservoir, Khon Kaen province in Thailand, and Intamat et al. [45] found As, Cr, Cd, Ni, Fe and Mn in the muscles of *Rasbora*
*tornieri* around the gold mine area of Loei province, Thailand.

### 4.3. Chromosome Study in H. macrolepidota

The results of this study revealed six types of chromosome abnormalities. The statistical results revealed that the number of cells with chromosome abnormalities, the total number of chromosome abnormalities and the percentages of chromosome abnormalities were significantly different (*p* < 0.05) between the *H**. macrolepidota* fish samples from the Nam Kok River near the gold-copper mine and those from the Nam Souang River. Furthermore, compared with bonylip barb fish (*Osteochilus vittatus*) at the same study site [15], *H**. macrolepidota* fish demonstrated fewer chromosomal effects in terms of the number of cells with chromosome abnormalities, the total number of chromosome abnormalities, and the percentages of chromosome abnormalities. These results could suggest that differences in metal toxicity consequences are species dependent.

Generally, potentially toxic elements depend on the fish species, the time of exposure, the concentrations of toxicants, the organ of exposure and excretion from the organs of each organism. Some scientists have studied chromosome abnormalities in aquatic animals around gold mine areas, wastewater canals, electronic waste and in in vitro experiments. Phoonaploy et al. [53] found five types of chromosome abnormalities in snakehead fish in reservoirs near industrial areas. Khamlerd et al. [46] reported that *Channa striata* affected by heavy metals (As, Pb, Cd, Cr, Cu, Fe, Zn, Mn and Ni) from the Bueng Jode reservoir in Thailand showed seven types of chromosome abnormalities. Sriuttha et al. [50] and Tengjaroenkul et al. [54] demonstrated six types and five types of chromosome abnormalities in *O**. niloticus* in a domestic wastewater canal and in an in vitro experiment, respectively. Neeratanaphan et al. [16] conducted a cytotoxic assessment of *Esomus metallicus* from a gold mine area in the Wang Saphung district of Loei Province in Thailand and found six types of chromosome abnormalities. In addition, research on *Monopterus albus* exposed to potentially toxic element contamination near a gold mine area found seven types of chromosome abnormalities [52]. The above cytotoxicity phenomena after Pb, As, Cd and Ni exposure in aquatic animals has been explained by several researchers. Pb may involve several indirect toxic mechanisms, such as oxidative damage, mitogenesis, DNA damage and alterations in gene transcription [55], and As could alter mitotic and replication indices, induce single strand DNA breaks and apoptosis, and inhibit DNA repair [56,57]. Cd can cause ploidy changes, oxidized bases, deletions and point mutations [58,59,60], and Ni can intervene in calcium-dependent enzyme activities, genetic expression, cell proliferation and DNA methylation [61,62]. From the negative effects indicated above, this study suggests that the potentially toxic elements accumulated in fish species can cause a decline in the fish population near a gold mine area.

### 4.4. Serum Biohemistry Study in H. macrolepidota

Measurements of the ALT and AST levels are valuable in the diagnosis of hepatic malfunctions as fish-metal health effects as well as the status of aquatic ecosystems. In this study, the AST and ALT of the fish from the study area near the Sepon gold-copper mine were significantly higher than those of the fish in the control area (*p*< 0.05), suggesting the occurrence of hepatic injury or liver impairment, likely due to potentially toxic element accumulation in this organ of *H**. macrolepidota,* as in other fish. Rao et al. [63] found an increase in the level of AST in the maternal tissue of *H**. fulvipes* as a response to the stress induced by potentially toxic elements. Asgah et al. [64] reported increases in the AST and ALT levels after exposure to Cd in the Nile tilapia *O. niloticus*. Naga et al. [65] revealed significant potentially toxic element consequences on AST and ALT in marine fish *Mugil seheli*. Furthermore, potentially toxic elements can negatively affect fish health in terms of protein, carbohydrate and lipid profiles [66]. Furthermore, compared with bonylip barb fish (*O*. *vittatus*) at the same study site [15], *H**. macrolepidota* fish demonstrated greater levels of both liver enzymes, particularly AST, which is directly related to the metal concentrations in the liver of *H**. macrolepidota* fish.

### 4.5. Histopathology Study in H. macrolepidota

The liver is a major organ involved in the accumulation and detoxification of toxic substances. Potentially toxic elements adversely affect the function of aquatic animal organs, causing changes in hepatic enzyme activities, extravasation of blood and necrosis of liver cells, fusion of gill lamellae, and genotoxicity [67,68,69,70]. Since the Nam kok River (as the study area located near the gold-copper mine) demonstrated higher levels of, and significantly different (*p* < 0.05), potentially toxic elements than the Nam Souang River (as a control area without mining activities), the abnormal liver cell structures in *H**. macrolepidota* from the study clearly revealed pathological lesions, including nuclear membrane degeneration, rough endoplasmic reticulum disintegration and abnormal cytoplasmic mitochondria. These changes imply that the potentially toxic element concentrations in *H**. macrolepidota* near the Sepon gold-copper mine could induce liver cell structural changes, as in the bonylip barb fish (*O*. *vittatus*) at the same study site [15]. The results were in accordance with several previous studies. Giari et al. [23] found mitochondrial and endoplasmic reticulum defects after exposure to Cd, and Dyk et al. [71] found that Cd exposure could damage the liver cell structures of *Oreochromis mossambicus*. Mishra and Mohanty [72] revealed that the occurrence of degenerative nuclei could be induced by metal free radicals. Vinodhini and Narayanan [73] reported that accumulated potentially toxic elements could cause cellular degeneration in the liver of *Cyprinus*
*carpio*.

## 5. Conclusions

The concentrations of As, Ba Cd, Cr, Cu, Fe, Mn, Ni, Pb, Se and Zn in water, sediment, and fish samples in the Nam Souang River near the agriculture and domestic areas, naturally enriched with some elements, were lower than those from the Nam Kok River near the gold-copper mine. The potentially toxic element concentrations (As, Ba, Cd, Cr, Cu, Fe, Mn, Ni, Zn and Pb) in the sediment were higher than those in the water samples, except for the undetected Se in the study area. Some potentially toxic element concentrations (Fe, Mn, Ni and Zn) in water exceeded the FAO, USA and WHO standards, while some potentially toxic elements (As, Cd, Cu and Zn) in sediment exceeded the EU and Thailand standards. Furthermore, the average concentrations of Cd, Cr and Mn in *H**. macrolepidota* from the study area exceeded the food quality standards of the EC and WHO. The toxicities of potentially toxic elements in *H**. macrolepidota* fish in the Nam Kok River near the Sepon gold-copper mine likely occurred due to their accumulation in fish cells, tissues, and organs, which can lead to abnormalities in chromosomes, changes in liver enzymes and alterations in the liver cell structures. The investigated information can apply to environmental and human health risk management and can contribute to the improvements of standards, regulations and policies regarding the mining industry in the Lao People’s Democratic Republic.

## Figures and Tables

**Figure 1 ijerph-18-05854-f001:**
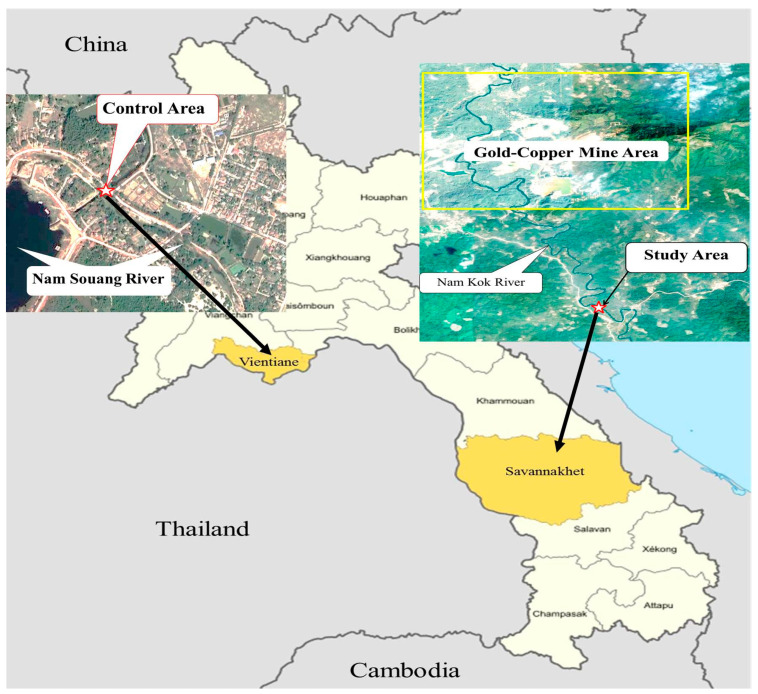
Geographical locations of the Nam Souang River as the control area (left) and the Nam Kok River near the Sepon gold-copper mine as the study area (right).

**Figure 2 ijerph-18-05854-f002:**
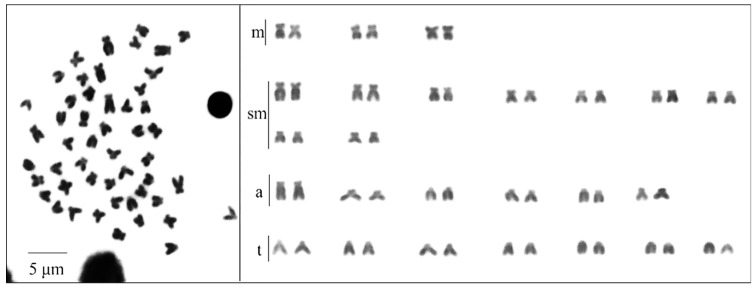
Karyotype of normal chromosome of *H**. macrolepidota* (2n = 50); metacentric (m), small metacentric (sm), acrocentric (a) and telocentric (t).

**Figure 3 ijerph-18-05854-f003:**
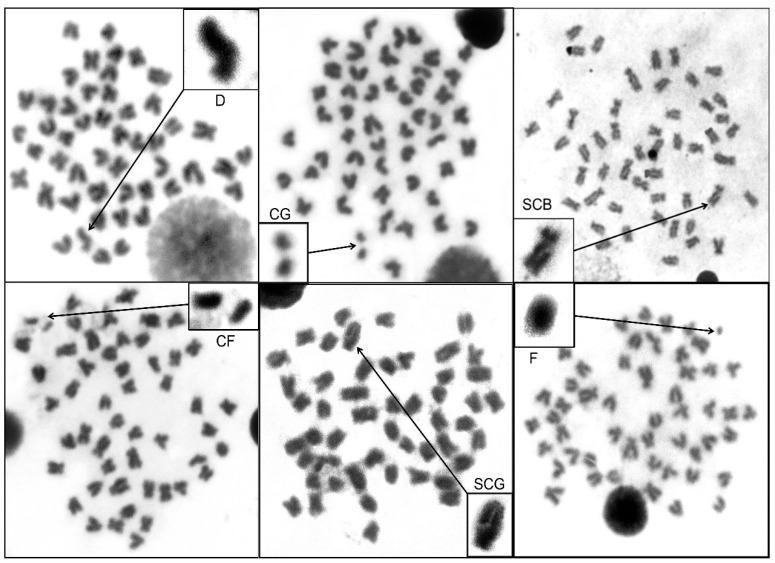
Six types of chromosome abnormalities in the metaphase spread cells of *H**. macrolepidota* (2n = 50); deletion (D), centromere gap (CG), single chromatid break (SCB), centric fermentation (CF), single chromatid gap (SCG) and fragmentation (F).

**Figure 4 ijerph-18-05854-f004:**
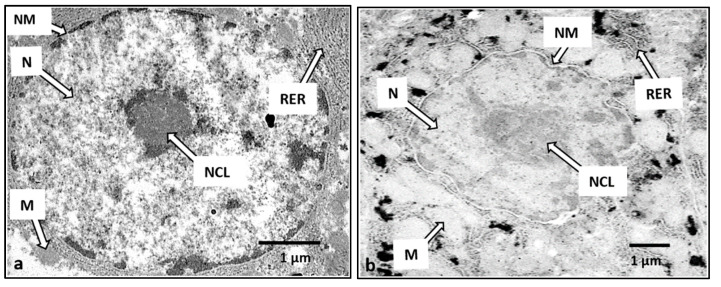
Electron micrograph of an *H. macrolepidota* liver cell. Nucleus (N), Nucleolus (NCL), Nuclear membrane (NM), Mitochondria (M), Rough endoplasmic reticulum (RER), Bar = 1 µm. Normal cell (**a**); Abnormal cell (**b**).

**Table 1 ijerph-18-05854-t001:** Analytical methods used for measurements of water quality parameters.

Water Quality Parameters	Analytical Methods
Dissolved oxygen	DO meter, Model 966, Mettler Toledo
pH	pH meter, Model EcoScan pH 5
Temperature	Eutech Thermometer
Total hardness, Carbonate hardness	Test kits, Chulalongkorn University, Thailand
Electrical conductivity	EC meter, Mettler Toledo
Nitrite, Nitrate, Ammonia	Titration methods

**Table 2 ijerph-18-05854-t002:** Water quality parameters in the Nam Kok River near the gold-copper mine (as the study area), and the Nam Souang River (as the control area).

Samples	Concentrations		*p*-Value
	Nam Kok River (Study Area)	Nam Souang River (Control Area)	
Temperature (°C)	23.65 ± 0.46	26.20 ± 0.67	0.009 *
DO (mg/L)	12.69 ± 0.77	7.96 ± 0.57	0.009 *
pH	8.08 ± 0.21	7.65 ± 0.34	0.007 *
TH (mg/L)	114.00 ± 5.48	94.00 ± 4.35	0.215
CH (mg/L)	11.00 ± 5.48	3.77 ± 0.48	0.154
Nitrite (mg/L)	0.03 ± 0.02	0.05 ± 0.02	0.118
Nitrate (mg/L)	7.43 ± 1.85	8.83 ± 3.67	0.062
Ammonia (mg/L)	0.02 ± 0.001	0.10 ± 0.01	0.065
EC (μs/cm)	443.40 ± 12.67	251.13 ± 6.24	0.207

* Pooled standard error of statistically significant difference (*p* < 0.05). Remarks: dissolved oxygen (DO), potential of hydrogen (pH), total hardness (TH), carbonate hardness (CH) and electrical conductivity (EC); (mean and standard deviation; n = 3).

**Table 3 ijerph-18-05854-t003:** Potentially toxic element concentrations in water samples (n = 9) of the Nam Kok River near the gold-copper mine (as the study area) and the Nam Souang River (as the control area).

Potentially Toxic Elements	Concentration in Water (mg/L)		*p*-Value	Standard
	Nam Kok River (Study Area)	Nam Souang River (Control Area)		
As	0.005 ± 0.002	0.001 ± 0.000	0.008 *	0.01 ^a^
Ba	5.581 ± 0.774	1.918 ± 1.281	0.009 *	0.05 ^g^
Cd	0.002 ± 0.001	0.001 ± 0.001	0.142	0.01 ^b^
Cr	0.017 ± 0.007	0.018 ± 0.015	0.465	0.05 ^d^
Cu	0.031 ± 0.006	0.010 ± 0.005	0.009 *	-
Fe	0.638 ± 0.126 **	0.910 ± 0.265 **	0.076	0.30 ^e^
Mn	0.109 ± 0.039 **	0.112 ± 0.016 **	0.347	0.05 ^f^
Ni	0.351 ± 0.436 **	0.017 ± 0.009	0.009 *	0.02 ^d^
Pb	0.038 ± 0.030	0.023 ± 0.019	0.347	0.05 ^b^
Se	0.007 ± 0.002	0.003 ± 0.003	0.047 *	-
Zn	3.526 ± 0.615 **	1.797 ± 0.978 **	0.028 *	0.20 ^c^

* Pooled standard error (*p*-value) of statistically significant difference (*p* < 0.05); ** Potentially toxic element concentration higher than standard reference value; Water standard references: ^a^ [24], ^b^ [25], ^c^ [26], ^d^ [27], ^e^ [28], ^f^ [29], ^g^ [30].

**Table 4 ijerph-18-05854-t004:** Potentially toxic element concentrations in sediment samples (n = 9) from the Nam Kok River near the gold-copper mine (as the study area) and the Nam Souang River (as the control area).

Potentially Toxic Elements	Concentration in Sediment (mg/kg)		*p*-Value	Standard
	Nam Kok River (Study Area)	Nam Souang River (Control Area)		
As	14.05 ± 7.30 **	0.03 ± 0.02	0.009 *	3.90 ^a^
Ba	429.59 ± 15.58	12.42 ± 16.07	0.009 *	-
Cd	2.35 ± 0.49 **	0.073 ± 0.06	0.009 *	0.16 ^b^
Cr	18.08 ± 5.96	0.48 ±0.22	0.009 *	45.50 ^b^
Cu	47.89 ± 16.55 **	0.42 ± 0.28	0.009 *	21.50 ^b^
Fe	13,198.38 ± 1861.08	214.94 ± 59.49	0.000 *	-
Mn	231.97 ± 95.91	2.95 ± 1.06	0.009 *	1800 ^a^
Ni	37.25 ± 8.06	1.89 ± 1.79	0.009 *	75.00 ^c^
Pb	50.50 ± 14.73	1.44 ± 0.93	0.009 *	300.00 ^c^
Se	ND	0.03 ± 0.03	-	-
Zn	343.45 ± 28.46 **	9.63 ± 8.23	0.009 *	300.00 ^c^

* Pooled standard error (*p*-value) of statistically significant difference (*p* < 0.05); ** Potentially toxic element concentration higher than standard reference value; sediment standard references: ^a^ [31], ^b^ [32], ^c^ [33].

**Table 5 ijerph-18-05854-t005:** Potentially toxic element concentrations in *H**. macrolepidota* samples (n = 9) from the Nam Kok River near the gold-copper mine (as the study area) and the Nam Souang River (as the control area).

Potentially Toxic Elements	Concentration (mg/kg)		*p*-Value	Standard
	Nam Kok River (Study Area)	Nam Souang River (Control Area)		
As	0.30 ± 0.10	0.17 ± 0.04	0.047 *	2.00 ^a^
Ba	0.89 ± 0.41	1.96 ± 0.52	0.016 *	-
Cd	0.21 ± 0.09 **	0.02 ± 0.001	0.009 *	0.05 ^b^
Cr	3.58 ± 0.54 **	1.67 ± 0.23	0.009 *	2.00 ^c^
Cu	2.20 ± 1.08	1.32 ± 0.22	0.028 *	10.00 ^c^
Fe	44.74 ± 19.21	39.33 ± 8.84	0.917	-
Mn	4.19 ± 0.81 **	2.14 ± 0.55 **	0.009 *	1.00 ^c^
Ni	1.70 ± 0.24	0.97 ± 0.11	0.009 *	-
Pb	ND	0.04 ± 0.04	-	0.20 ^d^
Se	1.21 ± 0.76	0.28 ± 0.08	0.009 *	-
Zn	79.25 ± 2.86	62.22 ± 15.95	0.047 *	-

* Pooled standard error (*p*-value) of statistically significant difference (*p* < 0.05); ** Potentially toxic element concentration higher than standard reference value; Standard references: ^a^ [34], ^b^ [35], ^c^ [36], ^d^ [37].

**Table 6 ijerph-18-05854-t006:** Total number of chromosome abnormalities, cell numbers with chromosome abnormalities and percentages of chromosome abnormalities in *H**. macrolepidota* from the Nam Kok River near the gold-copper mine (as the study area) and the Nam Souang River (as the control area) (median interquartile range, n = 9).

*H* *. macrolepidota*	Number of Chromosome Abnormalities						Total Number of CA	Cell Number with CA	Percentage of CA
	F	SCG	CG	D	SCB	CF			
Study area	0(1.5)	0(1.5)	21(20)	5(2.5)	0(0.5)	2(2.5)	30(19.5)	15(4)	30(8)
Total/Average *	3	3	127	22	1	15	171	73	29.20 *
Control area	0(0)	0(0)	6(2)	1(0.5)	0(0)	0(1)	8(2.5)	6(3.5)	12(7)
Total/Average *	0	0	30	6	0	3	39	27	10.80 *
*p*-value							0.009 **	0.009 **	0.009 **

* Average percentage of chromosome abnormality. ** Statistically significant difference (*p*-value < 0.05); Remarks: F = fragmentation, SCG = single chromatid gap, CG = centromere gap, D = deletion, SCB = single chromatid break, CF = centric fragment, CA = chromosome abnormalities.

**Table 7 ijerph-18-05854-t007:** Serum liver enzyme parameters of *H**. macrolepidota*.

Parameter	Nam Kok River (Study Area) (n = 9)	Nam Souang River (Control Area) (n = 9)
AST (IU/L)	66.34 ± 13.67 *	32.67 ± 4.34 *
ALT (IU/L)	63.38 ± 7.73 *	28.67 ± 2.27 *

* Significantly different in the same row (*p* < 0.05); Remarks: Aspartate aminotransferase (AST), Alanine aminotransferase (ALT).

## Data Availability

The data presented in this study are available on request from the corresponding author.

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
