# Peer review of "Cytogenetic, Serum Liver Enzymes and Liver Cell Pathology of the Hampala Barb Fish (Hampala macrolepidota) Affected by Toxic Elements in the Contaminated Nam Kok River near the Sepon Gold-Copper Mine, Lao PDR"

_ijerph, 2021, doi:10.3390/ijerph18115854_

Round 1
Reviewer 1 Report
Fieldwork was conducted in a river affected by mining activities, as well as a reference site. Water, sediment and native fish species were collected and analyzed from multiple angles. Field water chemistry information is provided, along with metal and metalloid concentrations. Toxicity effect at enzyme, chromosome, and tissue structure level was also studied for fish. The text in the results is a bit repetitive. It would be nice if DOI could be provided for references.
Major comments:
1. Title: "heavy metal": The term "heavy metal" has been widely used for decades, but more and more criticism was raised against using this term, such as this reference below. I suggest considering a different term as appropriate through the manuscript.
Reference: Pourret, O.; Hursthouse, A. It's Time to Replace the Term "Heavy Metals" with "Potentially Toxic Elements" When Reporting Environmental Research. Int. J. Environ. Res. Public Heal. 2019, 16, 4446, doi:10.3390/ijerph16224446.
2. L85, “The water samples were added with nitric acid, and the sediment samples were…” Please provide more details about how the samples were collected. For example, were water samples collected from the surface or integrated by depth? Were sediment samples dredged? Were water samples collected before sediment to avoid the concern of resuspension?
3. L116, “mg/kg”: The detection limits were expressed as mg/kg here. However, the concentrations in Table 3 – 5 were all in mg/L. Then it is more appropriate to provide detection limits in mg/L.
4. L191, “mg/kg”: Again, note that the unit in Table 4 is “mg/L”. Please verify—the same comment for L208 and Table 5.
5. L297, “WHO [27]”: Note that this reference is a drinking water guideline. I am not sure if it is a fair comparison unless there is evidence that river water could be consumed without treatment.
6. L338, “swamp eel”: In the discussion section, the authors used common names sometimes. I suggest the authors always provide the scientific name if possible.
Minor comments:
- L70, “reference area”: Is there any reference for using this area as a “control”?
-
L126, “0.075 KCl” : I guess the unit is “M”?
-
L138, “2,000 g”: Note that “rpm” was used in Section 2.5. Is there any specific reason for using different units?
-
L169, “0.1.918” I think it should be “1.918”.
-
Table 5: For Pb, why are three decimals for mean value and two for standard deviation?
- L231, “29.2%” and “10.80%”: why the number of decimals different?
-
L335, “Belangerrii”: Not sure what species it is, Johnius belangerii? Please verify and make it Italic.
- L380, “Buschini et al. [61] and Zhou et al. [62].” It seems this sentence is not complete.
Author Response
Reply to Reviewer 1 is attached file.

Reviewer 2 Report
In this manuscript, the author investigates the effect of heavy metal and metalloid on the chromosomes, liver enzyme activities and liver histopathological changes in H. macrolepidota fish. The experiment showed the heavy metal and metalloid affected the fish’s chromosomes and the enzyme activities quite significantly. However, several points need to be revised:
Major comments:
- The manuscript should be proof-read spell check to eliminate grammatical errors carefully and edit by a professional English editing.
- The author should provide images with higher resolution, e.g Figure 2 and Figure
- The manuscript mentioned the pathology of the liver; however, the pathological section of the liver tissue was not provided. The pathological section could ensure the abnormal anatomical of liver tissues.
- The immune gene analysis needs to be provided to prove immune activities and pathological changes.
- Please clarify the importance of these heavy metal and metalloid which were selected for this experiment in introduction or discussion section.
Minor comments:
- The font size, spacing, and alignment should be consistent in the whole manuscript, e.g. line 152, 164, 165, 226, etc.
- The author should describe the context in line 133 and 134 more clearly: Percentage of cell with chromosome abnormalities and percentage of chromosome abnormalities were calculated and analysed statistically.
- The concentration unit in table 2 was not labelled.
- The unit depicted in table 3, 4 and 5 is inconsistent with the units which stated in their respective descriptions. Line 190-191: 50.50±14.73 and 1.44±0.93; ND and 0.03±0.03, and 343.45±28.46 and 9.63±8.23 mg/kg, respectively (Table 4).
- Line 238 is missing a closing bracket, e.g. line 238.
- The abbreviations should provide in line 139, e.g. aspartate aminotransferase (AST) and alanine aminotransferase (ALT)
- The '*' symbol in Table 7, column 2 is superscripted. The '*' symbol should be check throughout the manuscript.
Author Response
Reply to Reviewer 2 is in attached file.

Reviewer 3 Report
The work is interesting and have relevant information, especially in a regional context. However, before publication some issues should be improved, especially:
- considering that the so called reference site may not be adequate for that purpose (see specific comments);
- In the results and discussion there are some text that could be removed without significant loss (see specific comments);
- English should be improved, especially in the discussion sectio (and also in the conclusions section).
Specific comments
Line 4: I think it is "in contaminated" not "contaminated in".
Abstract: Revise according to the suggestions made for the rest of the document. Mostly, bear in mind that the so called reference area may not be a reference area (see comments below).
Line 18: Not "the" but "a".
Lines 22-24: Refer all the parameters studied and let clear that those differences reflect higer values in the impacted area.
Lines 26-28: English needs revision.
Lines 55-57: If they are limited, some exist. If that is true, they should be referred here together with the explanation of main issues investigated there.
Lines 57-60: Make more clear that this is the objective of the present study.
Lines 69-70: But many other agriculture, industry and domestic activities can be responsible for important loads of the elements in study. Also, for some elements, regional waters can be naturally enriched. Therefore, that should be recognized here and the potential effects on the results of that limitation addressed in the discussion, including identifying for which elements this control area is probably more adequate and for which not. Also, because of that, the claimings should be more careful in the conclusions and abstract.
Lines 75-79: English needs revision.
Lines 82-87: Let clear what you meant by three replicates of fish samples and how they relate with the 10 individuals referred as analysed for different analyses. Also, according to what is exposed in the results section, the water and sediments have also 10 samples. revise thorougly this part.
Lines 89-91: Important to give statistical evidence that the fishes captured in both areas have no differences (or at least that they have minor differences) in size as the concentration of metals usually vary with the size of the specimens.
Lines 149-150: Clarify in the text which were really the tested hypotheses. Also, not clear how were those comparisons made. Did you use each specimen as an independent observation and compare both subsets with the test? Let it clear in the text.
Lines 150-154: Same comments as for the previous sentence.
Table 2: The tests presented in this table were not referred in the methods section as they should.
Lines 167-179: This paragraph could be largely reduced if becoming less descriptive of the values available in table 3 and more focused in highlighting the differences observed.
Line 183: Standard reference values for which purpose? Include.
Lines 186-195: Same comment as for the previous paragraph (167-179). Only the last five lines seem relevant.
Line 200: Standard references for which purpose? Include.
Lines 202-214: Same comment as for the previous two paragraphs (167-179 and 186-195). Only the last six lines seem relevant.
Line 219: Standard reference values for which purposes? Include.
Lines 225-229: English needs revision.
Lines 231: Percentages should have the same significant numbers.
Lines 238: This n=5 do not match with the value (10) referred in the methodology. Revise.
Lines 268-271: The first sentence is not necessary. The English of the second one should be revised, including also stating that what is higher are their concentrations.
Table 7: This n=3 do not match with the value (10) referred in the methodology. Revise.
Lines 276-278: English needs revison. Also, without comparisons between both areas, this information is of little use for this work. Revise.
Discussion: English of the discussion section is fairly substandard. So, it should be carefully revised.
Lines 296-297: Water standards for which purpose? Include.
Lines 298-301: The problem here is that, according to the values shown in table 3, most of these parameters also exceed those standard values in the supposed reference area. So, or the natural values for these parameters in this region are higher than the standards usually considered or the so called reference area is really not a good reference area regarding those parameters. In both cases, considering only these results it is not possible to be so assertve about the impacts in the water of the mining activities. So, revise this paragraph to low down the tone of the claims and to consider these alternatives (natural values and not adequate reference areas).
Line 306: Suggested is not the best term when statistically significant differences occur... Also, I would say that it would preferably to reiforce that those differences are really higher in the study area than in the so called reference area.
Line 308: Standard values for which purpose? Include.
Lines 317-318: Sentence dispensable.
Lines 321: Standard values for which purpose? Include.
Lines 332-344: I would probably stick here to those references about the same species or areas affected by the same problem (mining activities) and not trying to analyse other species in different contexts that add no interesting information to the issue being studied in this work.
Lines 348-349: These numbers are for which area?
Line 351: It is important to refer that they are not only significant different but higher in the study area than in the so called reference area.
Line 399: Important to make some comparisons here between both studied areas.
Conclusions: Revise the English and the content considering the comments made along the rest of the manuscript, especially considering the fact that the reference area could not be really a good reference area.
Author Response
Reply to Reviewer 3 is in attached file.

Round 2
Reviewer 1 Report
The manuscript has been improved, and I have a few minor comments as below:
1. L130 “HNO3”: What grade is the acid? Trace Metal Grade or ACS? Same comment for all the chemicals used, such as L137, 142, 165, ...
2. L134 “ICP-OES”: Please provide model information of the equipment.
3. L136 “exact weight of 1.0 g” It is hard to believe how to weigh exact 1.0 g; what balance was used?
4. L142 “fish muscle sample”: In general, metals in gill are more widely used to indicate the degree of pollution with metals, so Gill Surface Interaction Model was developed (and Biotic Ligand Model recently); while the muscle tissue is more for the consumption point of view.
5. L144 “on a hot plate for 1 h.” Is this step also at 60 oC?
6. L341-343: I am not sure if this statement is entirely accurate. If I look at Table 3, Fe, Mn, and Zn concentrations of the control area are higher than the standard.
7. L465-467: Note that in Table 3, the sample concentration in water is in mg/L, but in Table 4, the concentration in sediment is in mg/kg. I would assume the units are equal since the solvent is water. But please clarify so that the readers could compare concentrations in water vs. concentrations in sediment.
8. L469 “heavy metals” Please replace this term as the previous comment.
Author Response
Reply to reviewer 1 is attached file.

Reviewer 2 Report
The manuscript has been improved by adding more details. However, the author should provided a higher resolution images for Figure 3.In general, the author revised all of our comments on the previous submitted manuscript. Overall, this revised version is suitable for publication.
Author Response
Figure 3 was edited for a higher resolution.
Reviewer 3 Report
Generally, authors did not make a big wffort to cope with the reviewers suggestions and comments.
Especially, the comments 8), 29) and 38) of rewier 3 should be properly addressed along all manuscript. Not enough to replace "reference" by control...
Still not clear the dimension of the different samples and characteristics of replicates and how the statistical comparisons were made. Without that is imopossible to get a good idea of the merits of the work.
.
Author Response
Reply to reviewer 3 is attached file.
